# On the solutions of fractional differential equations with modified Mittag-Leffler kernel and Dirac Delta function: Analytical results and numerical simulations

**Mohammed Al-Refai**[1]*, **Dumitru Baleanu**[2,3], **A.K. Alomari**[4]

**1** Department of Mathematics, Yarmouk University, Irbed, Jordan, **2** Department of Computer Science and Mathematics, Lebanese American University, Beirut, Lebanon, **3** Institute of Space Sciences, Magurele-Bucharest, Romania, **4** Department of Mathematics, Faculty of Science, Islamic University of Madinah, Madinah, Saudi Arabia

* m_alrefai@yu.edu.jo

**Data availability statement:** All data are in the paper and/or supporting information files.

## Abstract

In this paper we define, for the first time, the modified fractional derivative with Mittage-Leffler kernel of Riemann–Liouville (R-L) type of arbitrary order $\delta > 0$. We derive the infinite series representations for the modified derivatives of R-L and Caputo types and present a relationship between them. We also investigate the modified derivatives for the Dirac delta functions, and study related fractional differential equations. Explicit solutions were presented for linear fractional differential equations with constant coefficients via the Laplace transform. A fractional model with the modified derivative is considered and numerical simulations were presented.

## Introduction

In recent years several fractional derivatives with nonsingular kernels were introduced, such as the Caputo-Fabrizio derivative which involves a kernel of exponential type [1] and the Atangana-Baleanu derivative which involves a Mittag-Leffler kernel [2]. These types of derivatives were utilized to model several dynamical systems and were extensively studied by several authors. It was noted that fractional differential equations (FDEs) with fractional derivatives involving nonsingular kernels admit a limitation in their solutions, see Lemmas 3.1, 3.2 and 4.1 in [3], Lemma 3.4 in [4], Proposition 2.1 in [5] and the extensive discussion in [6]. Mainly, extra necessary conditions were imposed to verify the solvability of such equations, which may effect the validity of such derivatives in modeling real life problems. To overcome this issue, recently Al-Refai and Baleanu in [7] have extended the Atangana-Baleanu derivative of Caputo type (ABC) to obtain the modified fractional derivative with Mittag-Leffler kernel of Caputo type $^M D_0^\alpha$ which involves a singular kernel. They proved that $(^M D_0^\alpha u)(0) \neq 0$, in general and as a result they presented a non-zero solution to the Cauchy problem

$$(^M D_0^\alpha u)(t) = \lambda u(t), \ t > 0, \ u(0) = u_0, \tag{0.1}$$

**Funding:** The author(s) received no specific funding for this work.

**Competing interests:** The authors have declared that no competing interests exist.

such that $0 < \alpha < 1$. Later on, the modified derivative of Caputo type was utilized to model several dynamical systems ([8–13]). A general hybrid coupled system of FDEs with the modified derivative was investigated in [14], which involves several dynamical systems in the literature as particular cases. Several numerical schemes were developed to tackle fractional order systems with the modified derivatives based on the Lagrange polynomials in [9], the Laplace Adomian decomposition method in [8], the Gaussian elimination combined with Taylor's expansion in [10] and on operational matrix approach in [15]. The basic theory of systems of FDEs with the modified derivative was discussed in [16], analytical solutions in closed forms were obtained for constant coefficients systems and a numerical scheme based on the collocation method was developed for nonlinear systems. The idea of obtaining the modified derivative was extended to other types of fractional derivatives with nonsingular kernels, see [17,18].

**Definition 0.1.** *[7] Let $u^{(n-1)} \in L^1(0, T)$, the modified derivative with Mittag-Leffler kernel of Caputo type and order $n - 1 < \delta < n$, is given by*

$$\left({}^M D_0^\delta u\right)(t) = \frac{B(\alpha)}{1 - \alpha}\left[ u^{(n-1)}(t) - E_\alpha(-\mu_\alpha t^\alpha) u^{(n-1)}(0) - \mu_\alpha\, t^{\alpha-1} * E_{\alpha,\alpha}(-\mu_\alpha\, t^\alpha) u^{(n-1)}(t) \right], \quad (0.2)$$

*where $\delta = \alpha + n - 1$, and $*$ denotes the convolution of two functions, and $B(\alpha)$ is a normalization function with $B(0) = B(T) = 1$.*

We stress on the fact that the kernel in the modified derivative $k(t) = t^{\alpha-1} E_{\alpha,\alpha}(-\mu_\alpha t^\alpha)$ possesses an integrable singularity at the origin. The integral operator corresponding to the modified derivative (0.2) is the same as the integral operator of the original ABC, and here we use the modified integral operator presented in [19]. Operators in the Caputo sense are widely used by researchers in both theoretical and application viewpoints, see [20,21] The aim of the current study is to address the challenges in solving fractional differential equations that involve fractional derivatives with non-singular kernels. To the best of our knowledge, this is the first study on fractional differential equations with modified derivative of R-L type. We remark that the modified Atangana-Baleanu derivative of the RL-type is an extension of the Atangana-Baleanu derivative, designed to operate in a broader space. This allows for the solvability of related fractional differential equations without the need for additional, unnecessary conditions, as demonstrated in Sect 4.

In this paper, we present the modified fractional derivative with Mittage-Leffler kernels of R-L type and study related problems. Section 1 is devoted to the extension of the modified derivatives. In Sect 2, we present infinite series representations for the modified derivatives of R-L and Caputo types and present the relation between them. In Sect 3, we present the modified derivative of the Dirac delta function and solve related FDEs analytically. An application is presented in Sect 4, with numerical simulations. Finally, some concluding remarks and suggestions for future work are presented in Sect 5.

The following known formulas will be used throughout the text [22,23].

$$\left(\mathcal{L}\, E_\alpha(\gamma t^\alpha)\right)(s) = \frac{s^{\alpha-1}}{s^\alpha - \gamma}, \quad \left|\frac{\gamma}{s^\alpha}\right| < 1 \tag{0.3}$$

$$\left(\mathcal{L}\, t^{\alpha-1} E_{\alpha,\alpha}(\gamma t^\alpha)\right)(s) = \frac{1}{s^\alpha - \gamma}, \quad \left|\frac{\gamma}{s^\alpha}\right| < 1, \tag{0.4}$$

$$\left(\mathcal{L}\, \left({}^M D_0^\alpha u\right)\right)(s) = \frac{B(\alpha)}{1 - \alpha}\frac{s^\alpha \mathcal{L}(u; s) - u(0)s^{\alpha-1}}{s^\alpha + \mu_\alpha},$$

$$\left|\frac{\mu_\alpha}{s^\alpha}\right| < 1. \tag{0.5}$$

$$\frac{d}{dt}\left(E_\alpha(-\mu_\alpha\,t^\alpha)\right) = -\mu_\alpha\,t^{\alpha-1}E_{\alpha,\alpha}(-\mu_\alpha\,t^\alpha). \tag{0.6}$$

$$\int_0^t (t-s)^{\alpha-1}E_{\alpha,\alpha}(-\mu_\alpha(t-s)^\alpha)u(s)ds = \sum_{k=0}^\infty (-\mu_\alpha)^k (I_0^{\alpha k+\alpha}u)(t), \tag{0.7}$$

where $E_{\alpha,\beta}(t) = \sum_{k=0}^\infty \frac{t^k}{\Gamma(\alpha k+\beta)}$, denotes the Mittag-Leffler function of two parameters, and $\mathcal{L}$ is the Laplace transform.

## 1 The modified fractional derivative with Mittage-Leffler kernel of RL-type

We start with the extension for $0 < \alpha < 1$. Let $u \in H^1(0, T)$, the Atangana-Baleanu derivative of RL-type is given by

$$({}_*D_0^\alpha u)(t) = \frac{B(\alpha)}{1-\alpha}\frac{d}{dt}\int_0^t E_\alpha(-\mu_\alpha(t-s)^\alpha)u(s)ds. \tag{1.1}$$

Let $U(t) = \int_0^t u(s)ds$, then integration by parts of the above equation and using (0.6) we have

$$({}_*D_0^\alpha u)(t) = \frac{B(\alpha)}{1-\alpha}\frac{d}{dt}\left(E_\alpha(-\mu_\alpha(t-s)^\alpha)U(s)\big|_0^t - \int_0^t \frac{d}{ds}\left(E_\alpha(-\mu_\alpha(t-s)^\alpha)\right)U(s)ds\right)$$

$$= \frac{B(\alpha)}{1-\alpha}\frac{d}{dt}\left(E_\alpha(0)U(t) - E_\alpha(-\mu_\alpha t^\alpha)U(0)\right.$$

$$\left. - \mu_\alpha\int_0^t (t-s)^{\alpha-1}E_{\alpha,\alpha}(-\mu_\alpha(t-s)^\alpha)U(s)ds\right).$$

Because $E_\alpha(0) = 1$, and $U(0) = 0$, we arrive at

$$({}_*D_0^\alpha u)(t) = \frac{B(\alpha)}{1-\alpha}\frac{d}{dt}\left(U(t) - \mu_\alpha\int_0^t (t-s)^{\alpha-1}E_{\alpha,\alpha}(-\mu_\alpha(t-s)^\alpha)U(s)ds\right)$$

$$= \frac{B(\alpha)}{1-\alpha}\left(u(t) - \mu_\alpha\frac{d}{dt}\int_0^t (t-s)^{\alpha-1}E_{\alpha,\alpha}(-\mu_\alpha(t-s)^\alpha)U(s)ds\right).$$

**Definition 1.1.** *Let $u \in L^1(0, T)$, the modified fractional derivative with Mittage-Leffler kernel of RL-type and order $0 < \alpha < 1$, is given by*

$$({}_*^M D_0^\alpha u)(t) = \frac{B(\alpha)}{1-\alpha}\left(u(t) - \mu_\alpha\frac{d}{dt}\int_0^t (t-s)^{\alpha-1}E_{\alpha,\alpha}(-\mu_\alpha(t-s)^\alpha)U(s)ds\right), \tag{1.2}$$

*where $U(t) = \int_0^t u(s)ds$.*

We apply the above approach to obtain the modified higher order derivatives of order $n - 1 < \delta < n, n \in \mathbb{N}$, $\delta = \alpha + n - 1$ and $0 < \alpha < 1$. Let $u^{(n-1)} \in H^1(0, T)$, we have

$$({}_*D_0^\delta u)(t) = \frac{B(\alpha)}{1-\alpha}\frac{d^n}{dt^n}\int_0^t E_\alpha(-\mu_\alpha(t-s)^\alpha)u(s)ds. \tag{1.3}$$

Integration by parts will lead to

$$({}_*D_0^\alpha u)(t) = \frac{B(\alpha)}{1-\alpha} \frac{d^n}{dt^n}\left(U(t) - \mu_\alpha \int_0^t (t-s)^{\alpha-1} E_{\alpha,\alpha}(-\mu_\alpha(t-s)^\alpha) U(s)\,ds\right)$$

$$= \frac{B(\alpha)}{1-\alpha}\left(u^{(n-1)}(t) - \mu_\alpha \frac{d^n}{dt^n}\int_0^t (t-s)^{\alpha-1} E_{\alpha,\alpha}(-\mu_\alpha(t-s)^\alpha) U(s)\,ds\right).$$

**Definition 1.2.** *For $n \;-\; 1 < \delta < n, n \in \mathbb{N},\; \delta = \alpha \;+\; n \;-\; 1,\; 0 < \alpha < 1$ and $u^{(n-1)} \in L^1(0,T)$, the modified fractional derivative with Mittage-Leffler kernel of RL-type and order $n \;-\; 1 < \delta < n$, is given by*

$$({}_*^M D_0^\delta u)(t) = \frac{B(\alpha)}{1-\alpha}\left(u^{(n-1)}(t) - \mu_\alpha \frac{d^n}{dt^n}\int_0^t (t-s)^{\alpha-1} E_{\alpha,\alpha}(-\mu_\alpha(t-s)^\alpha) U(s)\,ds\right). \qquad (1.4)$$

## 2 Infinite series representations

We derive infinite series representations of the modified derivatives in terms of the R-L fractional integral, and establish a relationship between them. We apply the following facts about the R-L fractional integral operator

$$(D^n I_0^\beta u)(t) = (I_0^{\beta-n} u)(t),\; \beta \in \mathbb{R}^+, n \in \mathbb{N}, \qquad (2.1)$$

$$(I_0^\beta D^n u)(t) = (I_0^{\beta-n} u)(t) - \sum_{j=0}^{n-1} \frac{t^{j+\beta-n}}{\Gamma(j+\beta-n+1)} u^{(j)}(0),\; \beta \in \mathbb{R}^+,\; n \in \mathbb{N}, \qquad (2.2)$$

where $D^n$ denotes the integer derivative of order $n$, and $I_0^\beta$ is the R-L fractional integral of order $\beta > 0$.

### 2.1 Modified fractional derivative with Mittage-Leffler kernel of RL-type

Using the results in Eqs (0.7) and (2.1) we arrive at the following infinite series representation of the modified fractional derivative with Mittage-Leffler kernel of RL-type

$$({}_*^M D_0^\delta u)(t) = \frac{B(\alpha)}{1-\alpha}\left(u^{(n-1)}(t) - \mu_\alpha \frac{d^n}{dt^n}\sum_{k=0}^\infty (-\mu_\alpha)^k (I_0^{\alpha k+\alpha} U)(t)\right),\; n-1 < \delta < n,$$

$$= \frac{B(\alpha)}{1-\alpha}\left(u^{(n-1)}(t) - \mu_\alpha \frac{d^n}{dt^n}\sum_{k=0}^\infty (-\mu_\alpha)^k (I_0^{\alpha k+\alpha+1} u)(t)\right)$$

$$= \frac{B(\alpha)}{1-\alpha}\left(u^{(n-1)}(t) - \mu_\alpha \sum_{k=0}^\infty (-\mu_\alpha)^k (I_0^{\alpha k-n+\alpha+1} u)(t)\right), \qquad (2.3)$$

and for $0 < \alpha < 1$,

$$({}_*^M D_0^\alpha u)(t) = \frac{B(\alpha)}{1-\alpha}\left(u(t) - \mu_\alpha \frac{d}{dt}\sum_{k=0}^\infty (-\mu_\alpha)^k (I_0^{\alpha k+\alpha+1} u)(t)\right)$$

$$= \frac{B(\alpha)}{1-\alpha}\left(u(t) - \mu_\alpha \sum_{k=0}^\infty (-\mu_\alpha)^k (I_0^{\alpha k+\alpha} u)(t)\right). \qquad (2.4)$$

## 2.2 Modified fractional derivative with Mittage-Leffler kernel of Caputo type

For $0 < \alpha < 1$, the infinite series representation of the modified derivative of Caputo type was derived in [7]. For arbitrary $n - 1 < \delta < n$, $n \in \mathbb{N}$, we have

$$({}^{M}D_0^{\delta}u)(t) = \frac{B(\alpha)}{1-\alpha}\left[u^{(n-1)}(t) - E_{\alpha}(-\mu_{\alpha}t^{\alpha})u^{(n-1)}(0) - \mu_{\alpha}t^{\alpha-1}E_{\alpha,\alpha}(-\mu_{\alpha}t^{\alpha}) * u^{(n-1)}(t)\right]. \quad (2.5)$$

Using the results in (0.7) and (2.2) it holds that

$$I_t(t) = t^{\alpha-1}E_{\alpha,\alpha}(-\mu_{\alpha}t^{\alpha}) * u^{(n-1)}(t) = \int_0^t (t-s)^{\alpha-1}E_{\alpha,\alpha}(-\mu_{\alpha}(t-s)^{\alpha})u^{(n-1)}(s)ds$$

$$= \sum_{k=0}^{\infty}(-\mu_{\alpha})^k(I_0^{\alpha k+\alpha}u^{(n-1)})(t)$$

$$= \sum_{k=0}^{\infty}(-\mu_{\alpha})^k\left((I_0^{\alpha k+\alpha-n+1}u)(t) - \sum_{j=0}^{n-2}\frac{t^{j+\alpha k+\alpha-n+1}}{\Gamma(j+\alpha k+\alpha-n+2)}u^{(j)}(0)\right)$$

$$= \sum_{k=0}^{\infty}(-\mu_{\alpha})^k(I_0^{\alpha k+\alpha-n+1}u)(t)$$

$$- \sum_{j=0}^{n-2}u^{(j)}(0)t^{j+\alpha-n+1}\sum_{k=0}^{\infty}(-\mu_{\alpha})^k\frac{t^{\alpha k}}{\Gamma(j+\alpha k+\alpha-n+2)}$$

$$= \sum_{k=0}^{\infty}(-\mu_{\alpha})^k(I_0^{\alpha k+\alpha-n+1}u)(t) - \sum_{j=0}^{n-2}u^{(j)}(0)t^{j+\alpha-n+1}E_{\alpha,j+\alpha-n+2}(-\mu_{\alpha}t^{\alpha})$$

$$= \sum_{k=0}^{\infty}(-\mu_{\alpha})^k(I_0^{\alpha k+\alpha-n+1}u)(t) - \sum_{j=0}^{n-1}u^{(j)}(0)t^{j+\alpha-n+1}E_{\alpha,j+\alpha-n+2}(-\mu_{\alpha}t^{\alpha})$$

$$+ u^{(n-1)}(0)t^{\alpha}E_{\alpha,\alpha+1}(-\mu_{\alpha}t^{\alpha}).$$

Because $t^{\alpha}E_{\alpha,\alpha+1}(-\mu_{\alpha}t^{\alpha}) = -\frac{1}{\mu_{\alpha}}\left(E_{\alpha}(-\mu_{\alpha}t^{\alpha}) - 1\right)$, we arrive at

$$I_t(t) = \sum_{k=0}^{\infty}(-\mu_{\alpha})^k(I_0^{\alpha k+\alpha-n+1}u)(t)$$

$$- \sum_{j=0}^{n-1}u^{(j)}(0)t^{j+\alpha-n+1}E_{\alpha,j+\alpha-n+2}(-\mu_{\alpha}t^{\alpha})$$

$$- \frac{u^{(n-1)}(0)}{\mu_{\alpha}}\left(E_{\alpha}(-\mu_{\alpha}t^{\alpha}) - 1\right). \quad (2.6)$$

Substituting the above result in Eq (2.5) will lead to

$$({}^{M}D_0^{\delta}u)(t) = \frac{B(\alpha)}{1-\alpha}\left[u^{(n-1)}(t) - u^{(n-1)}(0)\right.$$

$$+ \mu_{\alpha}\sum_{j=0}^{n-1}u^{(j)}(0)t^{j+\alpha-n+1}E_{\alpha,j+\alpha-n+2}(-\mu_{\alpha}t^{\alpha})$$

$$\left. - \mu_{\alpha}\sum_{k=0}^{\infty}(-\mu_{\alpha})^k(I_0^{\alpha k+\alpha-n+1}u)(t)\right]. \quad (2.7)$$

As a particular case of the above result and for $0 < \alpha < 1$, and $n = 1$, we have

$$
\begin{aligned}
\left({}^{M}D_0^\alpha u\right)(t) &= \frac{B(\alpha)}{1 - \alpha}\Bigg[ u(t) - u(0) + u(0)\mu_\alpha t^\alpha E_{\alpha,\alpha+1}(-\mu_\alpha t^\alpha) \\
&\quad - \mu_\alpha \sum_{k=0}^{\infty} (-\mu_\alpha)^k \left(I_0^{\alpha k + \alpha} u\right)(t) \Bigg] \\
&= \frac{B(\alpha)}{1 - \alpha}\Bigg[ u(t) - u(0)E_\alpha(-\mu_\alpha t^\alpha) - \mu_\alpha \sum_{k=0}^{\infty} (-\mu_\alpha)^k \left(I_0^{\alpha k + \alpha} u\right)(t) \Bigg]
\end{aligned}
\tag{2.8}
$$

which agrees with the result obtained in [7].

From the representation in Eqs (2.3) and (2.7) we arrive at the following relationship between the modified derivatives ${}^{M}_{*}D_0^\delta$ and ${}^{M}D_0^\delta$.

**Proposition 2.1.** *For $n - 1 < \delta < n, n \in \mathbb{N}$, $\delta = \alpha + n - 1$, $0 < \alpha < 1$, and $u^{(n-1)} \in L^1(0, T)$, it holds that*

$$
\begin{aligned}
\left({}^{M}_{*}D_0^\delta u\right)(t) &= \left({}^{M}D_0^\delta u\right)(t) + \frac{B(\alpha)}{1 - \alpha}\Bigg( u^{(n-1)}(0) \\
&\quad - \mu_\alpha \sum_{j=0}^{n-1} u^{(j)}(0) t^{j+\alpha-n+1} E_{\alpha, j+\alpha-n+2}(-\mu_\alpha t^\alpha) \Bigg).
\end{aligned}
\tag{2.9}
$$

As a particular case, and for $0 < \delta = \alpha < 1$, we have

$$
\begin{aligned}
\left({}^{M}_{*}D_0^\delta u\right)(t) &= \left({}^{M}D_0^\delta u\right)(t) + \frac{B(\alpha)}{1 - \alpha}\left( u(0) - \mu_\alpha u(0) t^\alpha E_{\alpha,\alpha+1}(-\mu_\alpha t^\alpha) \right) \\
&= \left({}^{M}D_0^\delta u\right)(t) + \frac{B(\alpha)}{1 - \alpha} u(0)\left( 1 - \mu_\alpha t^\alpha E_{\alpha,\alpha+1}(-\mu_\alpha t^\alpha) \right).
\end{aligned}
\tag{2.10}
$$

Because

$$
\begin{aligned}
t^\alpha E_{\alpha,\alpha+1}(-\mu_\alpha t^\alpha) &= t^\alpha \sum_{k=0}^{\infty} \frac{(-\mu_\alpha)^k t^{\alpha k}}{\Gamma(\alpha k + \alpha + 1)} = \sum_{k=0}^{\infty} \frac{(-\mu_\alpha)^k t^{\alpha k + \alpha}}{\Gamma(\alpha k + \alpha + 1)} \\
&= -\frac{1}{\mu_\alpha} \sum_{k=1}^{\infty} \frac{(-\mu_\alpha)^k t^{\alpha k}}{\Gamma(\alpha k + 1)} = -\frac{1}{\mu_\alpha}\left( E_\alpha(-\mu_\alpha t^\alpha) - 1 \right),
\end{aligned}
\tag{2.11}
$$

we arrive at

$$
\left({}^{M}_{*}D_0^\delta u\right)(t) = \left({}^{M}D_0^\delta u\right)(t) + \frac{B(\alpha)}{1 - \alpha} u(0) E_\alpha(-\mu_\alpha t^\alpha).
\tag{2.12}
$$

**Example 2.1.** *For the constant function $u(t) = c_0$, using the representation in Eq (2.3) we have*

$$
\begin{aligned}
\left({}^{M}_{*}D_0^\delta c_0\right)(t) &= -\mu_\alpha \frac{B(\alpha)}{1 - \alpha} \sum_{k=0}^{\infty} (-\mu_\alpha)^k \left(I_0^{\alpha k - n + \alpha + 1} c_0\right)(t) \\
&= -\mu_\alpha \frac{B(\alpha)}{1 - \alpha} \sum_{k=0}^{\infty} (-\mu_\alpha)^k \frac{c_0}{\Gamma(\alpha k - n + \alpha + 2)} t^{\alpha k - n + \alpha + 1} \\
&= c_0 \frac{B(\alpha)}{1 - \alpha} t^{1 - n} \sum_{k=0}^{\infty} (-\mu_\alpha)^{k+1} \frac{1}{\Gamma(\alpha k - n + \alpha + 2)} t^{\alpha k + \alpha}
\end{aligned}
$$

$$= c_0 \frac{B(\alpha)}{1-\alpha} t^{1-n} \sum_{k=1}^{\infty} (-\mu_\alpha)^k \frac{1}{\Gamma(\alpha k - n + 2)} t^{\alpha k}$$

$$= c_0 \frac{B(\alpha)}{1-\alpha} t^{1-n} \left( E_{\alpha, 2-n}(-\mu_\alpha t^\alpha) - \frac{1}{\Gamma(2-n)} \right). \tag{2.13}$$

## 3 Dirac delta function

We recall that the Dirac delta function has the following form, namely

$$\delta(t-b) = \begin{cases} 0, & t \neq b \\ \infty, & t = b, \end{cases} \qquad \int_{-\infty}^{\infty} \delta(t-b)\,dt = 1.$$

We remark here that there is no such real valued function with these properties, and the Dirac delta function is defined in the sense of distribution, or it can be captured by defining a special measure called the Dirac measure. The following properties hold true for the Dirac delta function

$$(A_1) \quad \int_{-\infty}^{\infty} h(t)\delta(t-b)\,dt = h(b), h \in L^1(\mathbb{R})$$

$$(A_2) \quad \mathcal{L}(\delta(t-b))(s) = e^{-bs},\ s > 0$$

$$(A_3) \quad \int_{-\infty}^{\infty} h(t)\frac{d^n}{dt^n}\delta(t-b)\,dt = (-1)^n \left(\frac{d^n}{dt^n} h\right)(b),\ h \in C^n(\mathbb{R}).$$

The above properties hold true for any interval $(c,d) \subset \mathbb{R}$, where $b \in (c,d)$. The R-L and Caputo derivatives of the Dirac delta were derived recently in [24,25] and related FDEs were studied. The modified fractional derivative with Mittage-Leffler kernel of Caputo type for the Dirac delta function is given by

$$({}^M D_0^\alpha \delta(t-b))(t) = \frac{B(\alpha)}{1-\alpha} \Bigg( \delta(t-b) - E_\alpha(-\mu_\alpha t^\alpha)\delta(t-b)(0)$$

$$- \mu_\alpha \int_0^t (t-s)^{\alpha-1} E_{\alpha,\alpha}(-\mu_\alpha(t-s)^\alpha)\delta(s-b)\,ds \Bigg).$$

Because $\delta(t-b)(0), b > 0$, and

$$\int_0^t (t-s)^{\alpha-1} E_{\alpha,\alpha}(-\mu_\alpha(t-s)^\alpha)\delta(s-b)\,ds = (t-b)^{\alpha-1} E_{\alpha,\alpha}(-\mu_\alpha(t-b)^\alpha)H(t-b),$$

where $H$ is the Heaviside function, then

$$({}^M D_0^\alpha \delta(t-b))(t) = \frac{B(\alpha)}{1-\alpha} \Bigg( \delta(t-b) - \mu_\alpha(t-b)^{\alpha-1} E_{\alpha,\alpha}(-\mu_\alpha(t-b)^\alpha)H(t-b) \Bigg). \tag{3.1}$$

Because $\delta(t-b)(0) = 0$, then

$$({}^M D_0^\alpha \delta(t-b))(t) = ({}^M_* D_0^\alpha \delta(t-b))(t). \tag{3.2}$$

Eq (3.1) leads to the fact that $\delta(t-b)(t)$ is a solution to the fractional initial value problem

$$(^M D_0^\alpha u)(t) - \frac{B(\alpha)}{1-\alpha} u(t) = -\frac{B(\alpha)}{1-\alpha} \mu_\alpha (t-b)^{\alpha-1} E_{\alpha,\alpha}(-\mu_\alpha (t-b)^\alpha) H(t-b)$$
$$u(0) = 0.$$

Using the result in Eq (0.5) we arrive at

$$(\mathcal{L}\,^M D_0^\alpha \,\delta(t-b))(s) = \frac{B(\alpha)}{1-\alpha} \frac{s^\alpha}{s^\alpha + \mu_\alpha} e^{-bs}, \ 0 < \alpha < 1, \ \ b > 0. \tag{3.3}$$

**Proposition 3.1.** *For* $0 < \alpha < 1$, *and* $\gamma_\alpha + \lambda \neq 0$, *the solution of the FDE*

$$(_*^M D_0^\alpha u)(t) + \lambda u(t) = f(t), \ t > 0, \tag{3.4}$$

*is given by*

$$u(t) = \frac{1}{\gamma_\alpha + \lambda} \left( f(t) + (\mu_\alpha - r_{\alpha,\lambda})((t^{\alpha-1} E_{\alpha,\alpha}(-r_{\alpha,\lambda} t^\alpha)) * f)(t) \right) \tag{3.5}$$

*where* $\gamma_\alpha = \frac{B(\alpha)}{1-\alpha}$, *and* $r_{\alpha,\lambda} = \frac{\lambda \mu_\alpha}{\gamma_\alpha + \lambda}$, *provided that* $\hat{f}(s) = (\mathcal{L}f)(s)$ *is well defined.*

*Proof*: Because

$$(\mathcal{L}\,_*^M D_0^\alpha u)(s) = (\mathcal{L}\,_* D_0^\alpha u)(s) = \frac{B(\alpha)}{1-\alpha} \frac{s^\alpha}{s^\alpha + \mu_\alpha} \hat{u}(s) = \gamma_\alpha \frac{s^\alpha}{s^\alpha + \mu_\alpha} \hat{u}(s),$$

applying the Laplace transform to (3.4) yields

$$\gamma_\alpha \frac{s^\alpha}{s^\alpha + \mu_\alpha} \hat{u}(s) + \lambda \hat{u}(s) = \left( \gamma_\alpha \frac{s^\alpha}{s^\alpha + \mu_\alpha} + \lambda \right) \hat{u}(s) = \hat{f}(s).$$

Direct calculations will lead to

$$\hat{u}(s) = \frac{s^\alpha + \mu_\alpha}{(\gamma_\alpha + \lambda)s^\alpha + \lambda \mu_\alpha} \hat{f}(s) = \frac{1}{\gamma_\alpha + \lambda} \frac{s^\alpha + \mu_\alpha}{s^\alpha + r_{\alpha,\lambda}} \hat{f}(s)$$
$$= \frac{1}{\gamma_\alpha + \lambda} \left( 1 + \frac{\mu_\alpha - r_{\alpha,\lambda}}{s^\alpha + r_{\alpha,\lambda}} \right) \hat{f}(s). \tag{3.6}$$

Applying the inverse Laplace transform we have

$$u(t) = \frac{1}{\gamma_\alpha + \lambda} \left( f(t) + (\mu_\alpha - r_{\alpha,\lambda})(t^{\alpha-1} E_{\alpha,\alpha}(-r_{\alpha,\lambda} t^\alpha) * f)(t) \right), \tag{3.7}$$

which completes the proof . □

**Proposition 3.2.** *For* $0 < \alpha < 1$, *and* $\gamma_\alpha + \lambda = 0$, *the solution of the FDE*

$$(_*^M D_0^\alpha u)(t) + \lambda u(t) = f(t), \ t > 0, \tag{3.8}$$

*is given by*

$$u(t) = -\frac{1}{\gamma_\alpha \mu_\alpha} \left( \frac{t^{-\alpha}}{\Gamma(1-\alpha)} * f' + f(0)\frac{t^{-\alpha}}{\Gamma(1-\alpha)} \right) - \frac{1}{\gamma_\alpha} f(t), \tag{3.9}$$

*where* $\gamma_\alpha = \frac{B(\alpha)}{1-\alpha}$, *and* $r_{\alpha,\lambda} = \frac{\lambda \mu_\alpha}{\gamma_\alpha + \lambda}$, *provided that* $f \in C^1(0, T)$ *and* $(\mathcal{L}f')(s)$ *is well defined.*

*Proof*: Applying the Laplace transform to (3.8) will lead to

$$\hat{f}(s) = (\mathcal{L}\,{}^M_* D^\alpha_0 u)(s) + \lambda \hat{u}(s) = \left( \gamma_\alpha \frac{s^\alpha}{s^\alpha + \mu_\alpha} + \lambda \right) \hat{u}(s)$$

$$= -\frac{\gamma_\alpha \mu_\alpha}{s^\alpha + \mu_\alpha} \hat{u}(s). \tag{3.10}$$

Thus,

$$\hat{u}(s) = -\frac{1}{\gamma_\alpha \mu_\alpha} (s^\alpha + \mu_\alpha)\hat{f}(s)$$

$$= -\frac{1}{\gamma_\alpha \mu_\alpha} \left( \frac{1}{s^{1-\alpha}} (s\hat{f}(s) - f(0)) + \frac{f(0)}{s^{1-\alpha}} \right) - \frac{1}{\gamma_\alpha}\hat{f}(s)$$

$$= -\frac{1}{\gamma_\alpha \mu_\alpha} \left( \frac{1}{s^{1-\alpha}} (\mathcal{L}f')(s) + \frac{f(0)}{s^{1-\alpha}} \right) - \frac{1}{\gamma_\alpha}\hat{f}(s). \tag{3.11}$$

Because $(\mathcal{L}\frac{t^{-\alpha}}{\Gamma(1-\alpha)})(s) = \frac{1}{s^{1-\alpha}}$, $s > 0$, the result follows by applying the inverse Laplace transform to (3.11). □

**Corollary 3.1.** *For* $0 < \alpha < 1$, *and* $\gamma_\alpha + \lambda \neq 0$, *the solution of the FDE*

$$({}^M D^\alpha_0 u)(t) + \lambda u(t) = f(t), \ t > 0, \tag{3.12}$$

*is given by*

$$u(t) = \frac{1}{\gamma_\alpha + \lambda} \left( g(t) + (\mu_\alpha - r_{\alpha,\lambda})((t^{\alpha-1}E_{\alpha,\alpha}(-r_{\alpha,\lambda}t^\alpha)) * g)(t) \right), \tag{3.13}$$

*where* $g(t) = f(t) + u(0)\gamma_\alpha E_\alpha(-\mu_\alpha t^\alpha)$.

*Proof*: From Eq (2.12) we have $({}^M D^\alpha_0 u)(t) = ({}^M_* D^\alpha_0 u)(t) - u(0)\gamma_\alpha E_\alpha(-\mu_\alpha t^\alpha)$, and the results follow by substituting the above result in Eq (3.4). □

Given that $(f * \delta(t-b))(t) = f(t-b)H(t-b)$, where $H$ is the Heaviside function, then the solution of

$$({}^M D^\alpha_0 u)(t) + \lambda u(t) = \delta(t-b), \ t > 0, \ u(0) = u_0 \tag{3.14}$$

is given by

$$u(t) = \begin{cases} \hat{u}, & t > 0 \\ u_0, & t = 0, \end{cases} \tag{3.15}$$

where,

$$\hat{u}(t) = \frac{1}{\gamma_\alpha + \lambda} \Bigg( \delta(t-b) + u(0)\gamma_\alpha E_\alpha(-\mu_\alpha t^\alpha)$$
$$+ (\mu_\alpha - r_{\alpha,\lambda})(t-b)^{\alpha-1} E_{\alpha,\alpha}(-r_{\alpha,\lambda}(t-b)^\alpha) H(t-b)$$
$$+ u(0)\gamma_\alpha(\mu_\alpha - r_{\alpha,\lambda}) \left( t^{\alpha-1} E_{\alpha,\alpha}(-\mu_\alpha t^\alpha) * E_\alpha(-r_{\alpha,\lambda} t^\alpha) \right)(t) \Bigg). \tag{3.16}$$

## 4 Numerical simulation

We consider the equation of the Resistor-Inductor circuit in the fractional case of the form

$$\left( {}^M D_0^\alpha I \right)(t) + R\, I(t) = \delta(t-1), \tag{4.1}$$

subject to the initial conditions $I(0) = a$. Here $I(t)$ represents the current flowing through the circuit, $v(t) = \delta(t-1)$ is the Dirac delta function input voltage, $L = 1$ is the inductance, and $R = 2$ is the resistance. The solution of the above system is given by Eq (3.13). The solution $I(t)$ within the range $0.92 \le \alpha \le 1$, and $I(0) = 0$ is depicted in Fig 1. Additionally, Fig 2 illustrates the solutions for $0.5 \le \alpha \le 0.9$ in increments of 0.1. Figs 3 and 4 present the solutions with $I(0) = 0.5$ for $0.92 \le \alpha \le 1$ in steps of 0.02, and for $0.5 \le \alpha \le 0.9$ in steps of 0.1 respectively. The curves intersect at $t = 0.571017$ and $t = 1.94714$, which enriches the dynamics of the system by varying $\alpha$, the order of the fractional derivative. All figures are plotted in the range of $t$ between 0 and 3, and using the first 1000 terms of the infinite sum. We remark here that if we replace the modified derivative of Caputo type with the original ABC-derivative, then the problem with the initial condition $I(0) = 0.5$ admits no solutions.

Memory effects are introduced into the system by the parameter $\alpha$, which controls the order of the fractional derivative and affects the damping and transient responsiveness of the current $I(t)$. While lower $\alpha$ values ($\alpha \approx 0.5$) show slower decay and stronger memory effects, higher $\alpha$ values ($\alpha \approx 1$) correlate to conventional integer-order dynamics with faster decay.

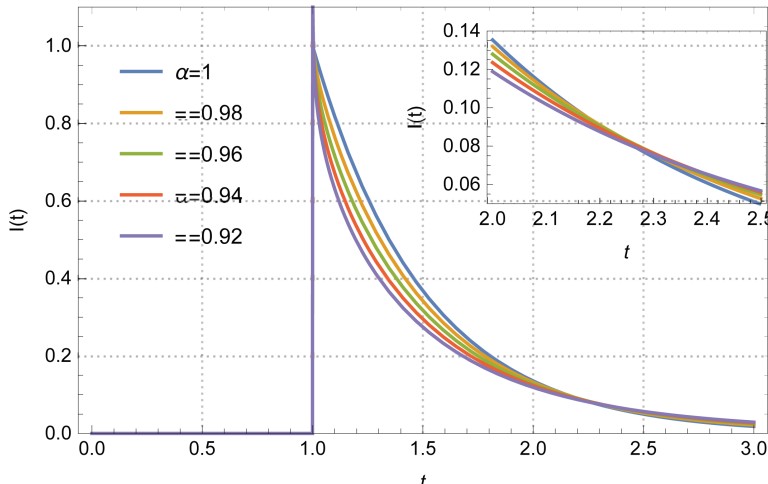

**Fig 1. The role of the fractional derivative on the solution behavior for Eq (4.1) with $I(0) = 0$ and several values of $\alpha$ between 0.92–1.0 by 0.02.**

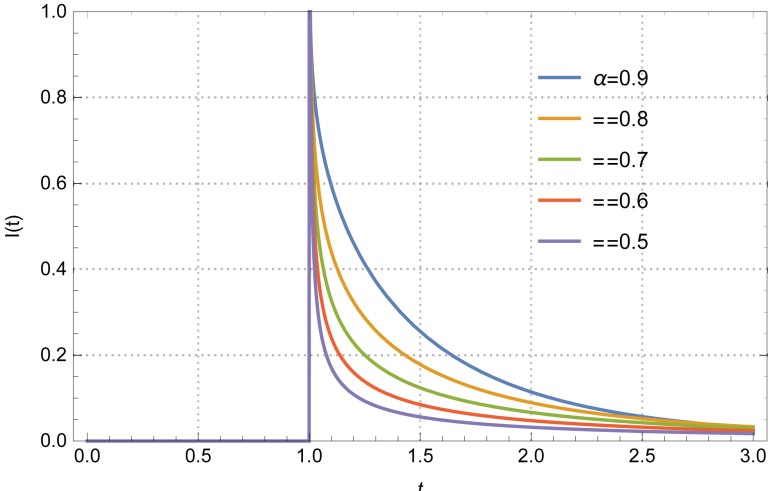

**Fig 2. The role of the fractional derivative on the solution behavior for Eq (4.1) with $I(0) = 0$ and several values of $\alpha$ between 0.5–0.9 by 0.1.**

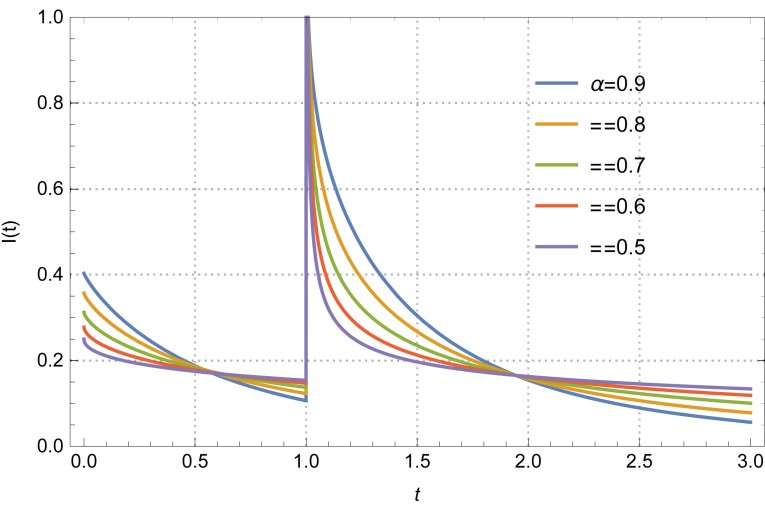

**Fig 3. The role of the fractional derivative on the solution behavior for Eq (4.1) with $I(0) = 0.5$ and several values of $\alpha$ between 0.92–1 by 0.02.**

The initial condition $I(0)$ is also important; when $I(0) = 0$, the current is driven only by the external input, but when $I(0) = 0.5$, an interaction between the external stimulus and the initial stored energy is added, which enhances the dynamics of the system. Furthermore, as the dynamics converges across various values of $\alpha$, the curve intersections at $t = 0.571017$ and $t = 1.94714$ demonstrate universal behavior and indicate crucial points of balance within the circuit.

## 5 Conclusion and further work

We have introduced the modified derivative with Mittage-Leffler kernel of R-L type, and developed the basic theory of related fractional differential equations. Infinite series

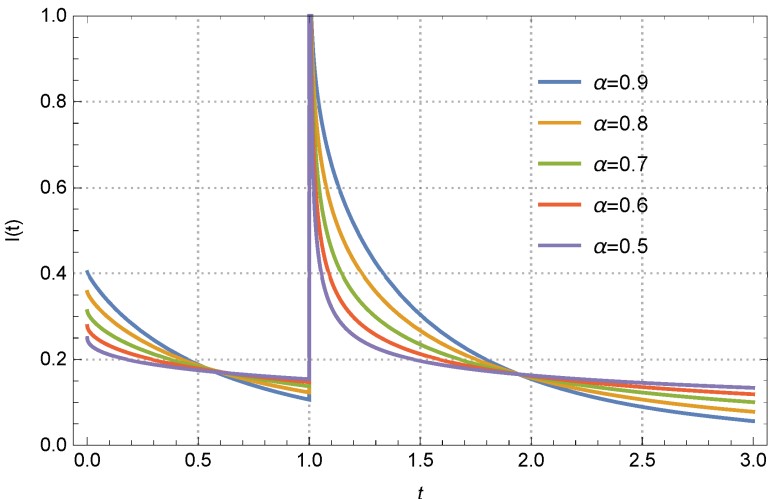

**Fig 4. The role of the fractional derivative on the solution behavior for Eq (4.1) with $I(0) = 0.5$ and several values of $\alpha$ between 0.5–0.9 by 0.1.**

representations of the modified derivatives of R-L and Caputo types were derived and implemented to obtain a closed formula for the relationship between the two derivatives. The modified derivative of the Dirac delta function is presented and related resistor-inductor model is discussed. The solutions of the presented model enrich the dynamics of the system and indicate the efficiency of implementing the modified derivative in modeling real life problems. In one side, the problem admits solutions without the need of imposing extra conditions, and as the fractional derivative approaches 1, the solution coincides with the solution obtained by solving the associated differential equation with integer derivative of order 1. In future work, we aim to implement the modified derivative in modeling various dynamical systems and to develop suitable numerical schemes for integrating these systems.

## Author contributions

**Conceptualization:** Mohammed Al-Refai, Dumitru Baleanu.

**Formal analysis:** Mohammed Al-Refai.

**Funding acquisition:** Dumitru Baleanu.

**Investigation:** Dumitru Baleanu, A.K. Alomari.

**Methodology:** Mohammed Al-Refai.

**Project administration:** Mohammed Al-Refai.

**Software:** A.K. Alomari.

**Supervision:** Dumitru Baleanu.

**Writing – original draft:** Mohammed Al-Refai.

**Writing – review & editing:** Mohammed Al-Refai, A.K. Alomari.

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
