## [Decision Letter · Decision Letter 0]

3 Jun 2025

PONE-D-24-46049On the solutions of fractional differential equations with modified Mittag-Leffler kernel and Dirac Delta function: Analytical results and numerical simulationsPLOS ONE

Dear Dr. Al-Refai,

Thank you for submitting your manuscript to PLOS ONE. After careful consideration, we feel that it has merit but does not fully meet PLOS ONE’s publication criteria as it currently stands. Therefore, we invite you to submit a revised version of the manuscript that addresses the points raised during the review process.Some reviewers suggested to cite the latest articles in the revised version. These reffered articles need not to be cited in the revise version. Please submit your revised manuscript by Feb 06 2025 11:59PM. If you will need more time than this to complete your revisions, please reply to this message or contact the journal office at plosone@plos.org. Please include the following items when submitting your revised manuscript:

We look forward to receiving your revised manuscript.

Kind regards,

Abuzar Ghaffari

Academic Editor

PLOS ONE

Journal Requirements: When submitting your revision, we need you to address these additional requirements. 1. Please ensure that your manuscript meets PLOS ONE's style requirements, including those for file naming. The PLOS ONE style templates can be found at https://journals.plos.org/plosone/s/file?id=wjVg/PLOSOne_formatting_sample_main_body.pdf and https://journals.plos.org/plosone/s/file?id=ba62/PLOSOne_formatting_sample_title_authors_affiliations.pdf 2. Please update your submission to use the PLOS LaTeX template. The template and more information on our requirements for LaTeX submissions can be found at http://journals.plos.org/plosone/s/latex. 3. Please provide a complete Data Availability Statement in the submission form, ensuring you include all necessary access information or a reason for why you are unable to make your data freely accessible. If your research concerns only data provided within your submission, please write "All data are in the manuscript and/or supporting information files" as your Data Availability Statement. 4. When completing the data availability statement of the submission form, you indicated that you will make your data available on acceptance. We strongly recommend all authors decide on a data sharing plan before acceptance, as the process can be lengthy and hold up publication timelines. Please note that, though access restrictions are acceptable now, your entire data will need to be made freely accessible if your manuscript is accepted for publication. This policy applies to all data except where public deposition would breach compliance with the protocol approved by your research ethics board. If you are unable to adhere to our open data policy, please kindly revise your statement to explain your reasoning and we will seek the editor's input on an exemption. Please be assured that, once you have provided your new statement, the assessment of your exemption will not hold up the peer review process.

**Additional Editor Comments:**

According to the reviewers, the article needs major revision to consider it for publication...

Comments from PLOS Editorial Office: We note that one or more reviewers has recommended that you cite specific previously published works. As always, we recommend that you please review and evaluate the requested works to determine whether they are relevant and should be cited. It is not a requirement to cite these works. We appreciate your attention to this request.

Reviewers' comments:

Reviewer's Responses to Questions

**Comments to the Author**

1. Is the manuscript technically sound, and do the data support the conclusions?

Reviewer #1: Yes

Reviewer #2: Yes

2. Has the statistical analysis been performed appropriately and rigorously? 

Reviewer #1: Yes

Reviewer #2: Yes

3. Have the authors made all data underlying the findings in their manuscript fully available?

Reviewer #1: Yes

Reviewer #2: Yes

4. Is the manuscript presented in an intelligible fashion and written in standard English?

Reviewer #1: Yes

Reviewer #2: Yes

5. Review Comments to the Author

Reviewer #1: Authors examined “Authors examined “On the solutions of fractional differential equations with modified Mittag-Leffler kernel and Dirac Delta function: Analytical results and numerical simulations.” The paper has potential to be published in this journal after following major revisions.

1.What are basic gaps which are filled in this research study?

2.Governing equation are note referenced. Reference each equation properly?

3.Please show the novelty and research gap at the end of introduction section by raising some questions which are to be addressed?

1.In the introduction references are little in number, increase it by adding more relevant studies.

2.Is it the original or relevant topic to the field, whose gaps remain to be filled?

3.How is this study compared to the previously published? Which issues are addressed in this which are not in the previously?

4.Is the methodology used here is new developed or the extended the previous one? If it is improved, then what are the basic improvements have been carried out?

5.Try to add some more relevant references to enrich the study literature for clear justification.

6.Authors are advised to explain the graph with physical reasoning about the parameters and properties in which the variations are being observed.

7.What is the main question addressed by the research?

8. What does it add to the subject area compared with other published

material?

9.What specific improvements should the authors consider regarding the

methodology? What further controls should be considered?

10.Are the conclusions consistent with the evidence and arguments presented

and do they address the main question posed?. Improve it.

11.What ranges of all parameters should be given? In the caption mention all of the involved pare metric values for which graphs are plotted, it will help to the reader compare the parametric effects for considered values of the parameters.

12.Write the future works in this studies.

Reviewer #2: Review Report

Title: On the Solutions of Fractional Differential Equations with Modified Mittag-Leffler Kernel and Dirac Delta Function: Analytical Results and Numerical Simulations

Overall Assessment

The paper addresses the solutions of fractional differential equations (FDEs) incorporating a modified Mittag-Leffler kernel and the Dirac delta function. This topic is both relevant and timely, as fractional calculus continues to find applications in modeling memory and hereditary effects in various physical systems. The use of the modified Mittag-Leffler kernel and the interplay with the Dirac delta function offer an interesting and novel perspective that could expand the applicability of fractional calculus.

While the manuscript contains valuable contributions, there are areas requiring improvement in terms of clarity, organization, and rigor. Below, I provide detailed comments and suggestions.

1.To enhance the clarity and accessibility of the proposed methodology, I suggest including a flowchart that outlines the key steps of the approach.

2.Highlight how the proposed method handles challenges like singularities or boundary conditions.

3.The manuscript includes only one numerical example, which, while illustrative, limits the depth of validation for the proposed methodology. To strengthen the results and provide a more comprehensive evaluation, I recommend the authors to compare the numerical example with solutions obtained using another established method from the literature, such as finite difference, finite element, or other spectral methods. Highlight the differences in accuracy, convergence rates, and computational efficiency between the proposed method and the chosen benchmark method.

4.The authors should consider citing additional relevant works in the field. Specifically

https://doi.org/10.1088/1402-4896/ac1ccf; https://doi.org/10.1155/2021/6662808; https://doi.org/10.1155/2020/1274251

6. PLOS authors have the option to publish the peer review history of their article (what does this mean?). If published, this will include your full peer review and any attached files.

Reviewer #1: No

Reviewer #2: No

---

## [Author Response · Author response to Decision Letter 1]

24 Jan 2025

We have done the necessary changes based on the reviewer's reports

---

## [Decision Letter · Decision Letter 1]

21 May 2025

On the solutions of fractional differential equations with modified Mittag-Leffler kernel and Dirac Delta function: Analytical results and numerical simulations

PONE-D-24-46049R1

Dear Dr. Al-Refai,

We’re pleased to inform you that your manuscript has been judged scientifically suitable for publication and will be formally accepted for publication once it meets all outstanding technical requirements.

Kind regards,

Mahmoud H. DarAssi, Ph.D

Academic Editor

PLOS ONE

Additional Editor Comments (optional):

Reviewers' comments:

Reviewer's Responses to Questions

**Comments to the Author**

1. If the authors have adequately addressed your comments raised in a previous round of review and you feel that this manuscript is now acceptable for publication, you may indicate that here to bypass the “Comments to the Author” section, enter your conflict of interest statement in the “Confidential to Editor” section, and submit your "Accept" recommendation.

Reviewer #1: All comments have been addressed

Reviewer #2: All comments have been addressed

2. Is the manuscript technically sound, and do the data support the conclusions?

Reviewer #1: Yes

Reviewer #2: Yes

3. Has the statistical analysis been performed appropriately and rigorously? 

Reviewer #1: Yes

Reviewer #2: N/A

4. Have the authors made all data underlying the findings in their manuscript fully available?

Reviewer #1: Yes

Reviewer #2: Yes

5. Is the manuscript presented in an intelligible fashion and written in standard English?

Reviewer #1: Yes

Reviewer #2: Yes

6. Review Comments to the Author

Reviewer #1: The authors have revised the manuscript properly by incorporating all of the corrections. Now the article is accepted for publication in its current form.

Reviewer #2: The authors have addressed all the comments raised during the review process. I find the revised manuscript acceptable for publication in its current form.

7. PLOS authors have the option to publish the peer review history of their article (what does this mean?). If published, this will include your full peer review and any attached files.

Reviewer #1: No

Reviewer #2: No

---

## [Editor Report · Acceptance letter]

PONE-D-24-46049R1

PLOS ONE

Dear Dr. Al-Refai,

I'm pleased to inform you that your manuscript has been deemed suitable for publication in PLOS ONE. Congratulations! Your manuscript is now being handed over to our production team.

Kind regards,

on behalf of

Dr. Mahmoud H. DarAssi

Academic Editor

PLOS ONE